# Stereoselective assembly of *C*-oligosaccharides via modular difunctionalization of glycals

Ya-Nan Ding[1], Mei-Ze Xu[1], Yan-Chong Huang[1], Lutz Ackermann [2] ✉, Xiangtao Kong [3] ✉, Xue-Yuan Liu[1] ✉ & Yong-Min Liang [1] ✉

C-oligosaccharides are found in natural products and drug molecules. Despite the considerable progress made during the last decades, modular and stereoselective synthesis of C-oligosaccharides continues to be challenging and underdeveloped compared to the synthesis technology of O-oligosaccharides. Herein, we design a distinct strategy for the stereoselective and efficient synthesis of C-oligosaccharides via palladium-catalyzed nondirected C1−H glycosylation/C2-alkenylation, cyanation, and alkynylation of 2-iodoglycals with glycosyl chloride donors while realizing the difunctionalization of 2-iodoglycals. The catalysis approach tolerates various functional groups, including derivatives of marketed drugs and natural products. Notably, the obtained C-oligosaccharides can be further transformed into various C-glycosides while fully conserving the stereochemistry. The results of density functional theory (DFT) calculations support oxidative addition mechanism of alkenyl-norbornyl-palladacycle (ANP) intermediate with α-mannofuranose chloride and the high stereoselectivity of glycosylation is due to steric hindrance.

Transition metal-catalyzed C−H glycosylation has surfaced as a powerful tool for synthesizing C-glycosides[1–6]. Iodoglycals can be used as both a useful glycosyl donor in C−H glycosidation reactions[7–13] and as readily available building blocks in functionalization reactions, which have long been of interest to chemists. Monofunctionalization of 2-iodoglycals via palladium-catalyzed cross-couplings can thus be considered as the most reliable strategy. For example, monofunctionalization typically involves palladium-catalyzed Heck[7], Stille[14], Suzuki–Miyaura[15], Sonogashira[16,17], aminocarbonylation[18], alkenylation[19], alkylation[20], among others[21,22] (Figs. 1, (1), left). Despite these indisputable advances, key challenges remain. First, all studies have as of yet focused on single cross-coupling reactions for the synthesis of 2-C-branched sugars, while difunctionalizations of 2-iodoglycals have not been reported. Second, the C−H

activation/glycosylation is highly promising[23–26]. In 2021, Chen's group reported a strategy for the stereoselective synthesis of C-vinyl glycosides via the palladium-catalyzed directed C-H glycosylation of olefins, but has been limited to the directed synthesis C-vinyl glycosides[27] (Figs. 1, (2)). In contrast, we herein describe the palladium-catalyzed C1−H glycosylation and C2-alkenylation of 2-iodoglycals through vicinal-difunctionalization without additional directing groups (Fig. 1, (1), right).

Compared with O-oligosaccharides, C-oligosaccharides have remarkable stability in the process of enzyme and chemical hydrolysis and are also widely present in natural products and drug molecules. For example, marine natural products (maitotoxin)[28] and drug molecules (dodecodiulose and tunicamine)[29,30] featuring interglycosidic C-linkages (Fig. 2, (1)). Therefore, glycosidic C−C bond formation

[1]State Key Laboratory of Applied Organic Chemistry and College of Chemistry and Chemical Engineering, Lanzhou University, 222 South Tianshui Road, 730000 Lanzhou, Gansu Province, China. [2]Institut für Organische und Biomolekulare Chemie and Wöhler-Research Institute for Sustainable Chemistry (WISCh), Georg-August-Universität, Tammannstrasse 2, 37077 Göttingen, Germany. [3]Henan Key Laboratory of New Optoelectronic Functional Materials, College of Chemistry and Chemical Engineering, Anyang Normal University, 455000 Anyang, China. ✉e-mail: lutz.Ackermann@chemie.uni-goettingen.de; kongxt@aynu.edu.cn; liuxuey@lzu.edu.cn; liangym@lzu.edu.cn

strategies are valuable for enriching the diversity of viable oligosaccharides[31]. In the last three decades, considerable progress has been made in the synthesis of C-oligosaccharides, among others, through ring-closing alkene metathesis[32], a Ramberg-Bäcklund approach[33] or Stille-type cross-coupling[34]. However, these approaches largely rely on highly reactive functionalized coupling partners, specifically designed glycosyl donors, delicate operating conditions, and multistep reaction sequences. In addition, the methods for assembling C-disaccharides that involve directly connecting the anomeric carbon and anomeric carbon are rare. This is most probably

attributed to the fact that connecting two sugar units C1−C1 together requires anomer carbon polarity reversal. This challenge has been overcome via Stille couplings[35] of glycals (Fig. 2, (2)) and the radical−radical homocouplings[36,37] (Fig. 2, (3)) of glycosides, but these reactions form dimeric products and anomeric mixtures with poor diastereoselectivity and have limited substrate scope. In 2022, the Ackermann group reported the Pd-catalyzed aminoquinoline-directed C($sp^3$)−H glycosylation of glycosides using 1-iodoglycals donors[12] (Fig. 2, (4)). This strategy has been proven to be effective, widely applicable, and diastereoselective, highlighting the important prospects of C-oligosaccharides. Despite these formidable advances in directed C−H glycosylation to synthesize C-oligosaccharides, the development of DG (directing group)-free C−H glycosylation has proven elusive (Fig. 2, (5)).

Over the last two decades, the Catellani-type reaction has allowed for highly efficient molecular polyfunctionalization[38–55], which has also provided us with a strategy for the undirected synthesis of C-oligosaccharides. Notably, in contrast to arene-Catellani reactions: (1) the π preactivity of alkene substrates enhances their reactivity and typically can lead to undesired cyclopropanation, (2) the preactivity of alkene substrates might more strongly interfere with the C−H cleavage process, posing a challenge to reaction development[27], (3) palladium-catalyzed nonstereoselective alkenyl Catellani reactions have been successful when using primary alkyl halides, their application in the synthesis of complex secondary C-alkenyl glycosides remains elusive.

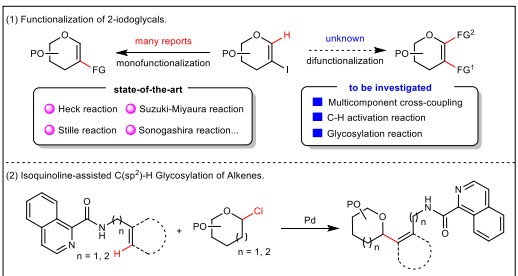

**Fig. 1 | Functionalization of 2-iodoglycals and syntheses of vinyl C-glycosyl. (1)** Functionalization of 2-iodoglycals. (**2**) Isoquinoline-assisted C($sp^2$)−H Glycosylation of Alkenes. FG functional group, PO protecting group.

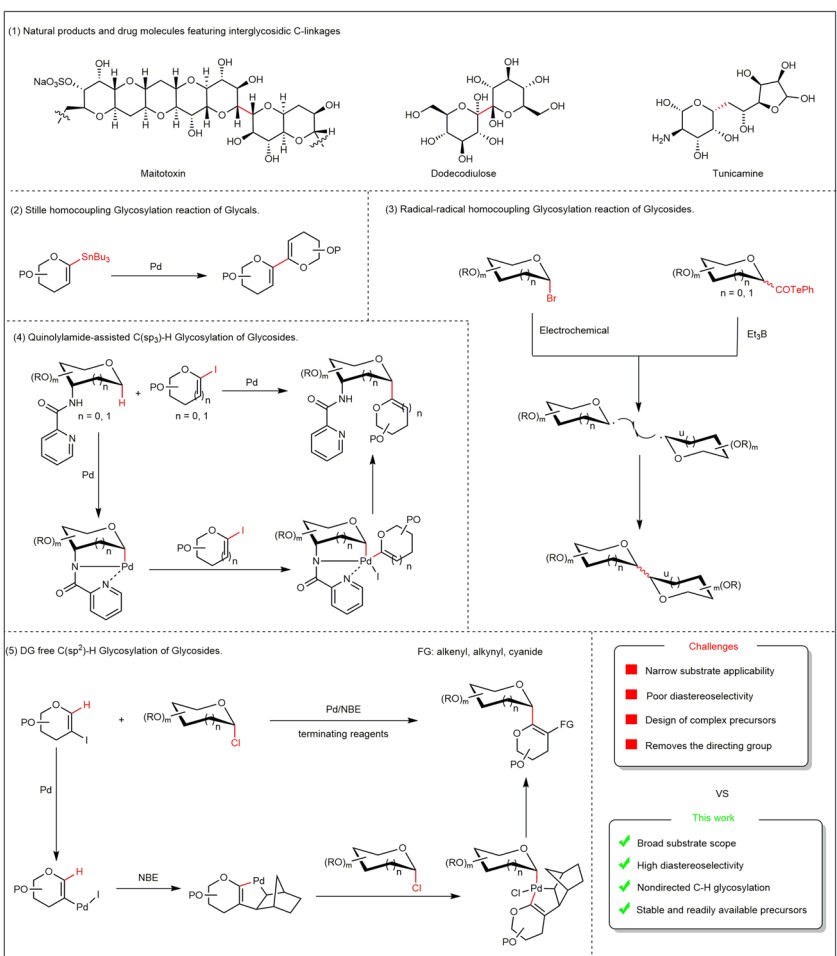

**Fig. 2 | Background of current work and reaction design. (1)** Natural products and drug molecules featuring interglycosidic C-linkages. (**2**) Synthesis of C-disaccharides via Stille homocoupling glycosylation reaction of glycosides. (**3**) Synthesis of C-disaccharides via radical-radical homocoupling glycosylation reaction of glycosides. (**4**) Quinolylamide-assisted C($sp^3$)−H glycosylation of glycosides. (**5**) Synthesis of C-disaccharides via nondirected C($sp^2$)−H glycosylation in a modular and stereoselective manner. FG functional group, PO and RO protecting group, NBE norbornene, DG directing group, Et₃B triethylborane.

**Table 1 | Optimization of reaction conditions[a]**

| Entry | Deviation from the standard conditions | Yield of 4a (%)[b] | Yield of 4a' (%)[b] |
|---|---|---|---|
| 1 | None | 64 | <5 |
| 2 | $K_2CO_3$ instead of $Cs_2CO_3$ | <10 | Trace |
| 3 | $Pd(OAc)_2$ instead of $Pd(PhCN)_2Cl_2$ | 57 | 8 |
| 4 | Toluene instead of 1,4-dioxane | 52 | 12 |
| 5 | DMF instead of 1,4-dioxane | Trace | 30 |
| 6 | $PPh_3$ instead of $L_1$ | 13 | 12 |
| 7 | TFP instead of $L_1$ | 22 | 18 |
| 8 | No $Pd(PhCN)_2Cl_2$ | 0 | 0 |
| 9 | 80 °C | 35 | <10 |
| 10 | 120 °C | 28 | <10 |
| 11 | No $L_1$ | 0 | 62 |
| 12 | No $N_8$ | 0 | 90 |
| 13 | No base | 0 | 0 |

*DMF* dimethylformamide, *TFP* tri(2-furyl)phosphine, *Ph-DavePhos* 2-Diphenylphosphino-2'-(N,N-dimethylamino)biphenyl.

[a]Reaction conditions: **1a** (0.15 mmol), **2a** (0.20 mmol), **3a** (0.10 mmol), $Pd(PhCN)_2Cl_2$ (0.01 mmol), $L_1$ (0.02 mmol), $N_8$ (0.20 mmol), $Cs_2CO_3$ (0.20 mmol), 100 °C, and 16 h.
[b]Isolated yield.

This is expected given the considerable challenges posed by multifunctional, sterically hindered sugar substrates and the additional need to control the diastereoselectivity of $C_{(alkenyl)}–C_{(anomeric)}$ bond formation. Following our continuous interest in Catellani-type reactions and carbohydrate chemistry[2,48,49,56–58], we questioned whether modular and stereoselective assembly of C-oligosaccharides via modular difunctionalization of glycals.

Here, we show the Catellani-type C–H glycosylation of 2-iodoglycals to afford C-oligosaccharides with excellent diastereoselectivities.

## Results

As shown in Table 1, we initiated our study by exploring the conditions for the envisioned palladium/norbornene (NBE)-enabled C–H glycosylation using 2-iodoglycal **1a**, glycosyl chloride donor **2a**, and ethyl acrylate **3a**. Interestingly, the model reaction afforded the C-disaccharide **4a** in 64% isolated yield with exclusive diastereoselectivity (entry 1). Compound **4a'** was obtained as byproduct via direct Mizoroki-Heck coupling, which can be inhibited by screening ligands and NBEs. $K_2CO_3$ was used to obtain the desired product in low yield (entry 2). $Pd(PhCN)_2Cl_2$ was found to be slightly more efficient than $Pd(OAc)_2$ (entry 3). Subsequently, we examined dimethylformamide (DMF) and toluene as solvents and found that the reaction could not proceed efficiently in polar reaction media (entries 4 and 5). Two experiments at high temperature (120 °C) and low temperature (80 °C) did not obtain higher yields (entries 9–10). Control experiments were then conducted to understand the role of each reaction component.

Unsurprisingly, no product was detected without palladium, ligand, NBE, or the base. (entries 8, 11–13). Dong had previously reported that the use of structurally modified norbornenes (smNBE) and Buchwald's Ph-DavePhos ($L_1$) increases the yield of the products. To our delight, we found that the use of Ph-DavePhos increased the yield of **4a** (entries 6 and 7 and $L_2$–$L_5$). Notably, the desired product was not formed when smNBE $N_4$ was used, whereas the yield of the direct Heck coupling product was >90%. We conclude that the possible reasons for this observation could be that smNBE forms a spatially crowded ANP and the glycosyl chloride with large steric hindrance cannot undergo oxidative addition, leading to the failure of this domino process. Therefore, we chose simple commercial norbornene to further probe our strategy. Among the investigated norbornenes, $N_8$ was identified as being optimal ($N_1$–$N_8$). We also preliminarily explained the possible reasons why $N_8$ is optimal through density functional theory (DFT) calculations (Supplementary Fig. 5).

### Substrate scope

After having established the optimized C–H glycosylation reaction conditions, various 2-iodoglycals, olefins, and glycosyl chloride donors were independently explored to examine the universality of the method to synthesize C-oligosaccharides (Fig. 3). Notably, excellent diastereoselectivity was observed in all cases, and the catalytic system was also compatible with different substitution pattern. The connectivity of the newly formed C-glycosidic bonds for each type of sugar was unambiguously verified via two-dimensional nuclear magnetic resonance spectroscopy. Various olefin terminating reagents,

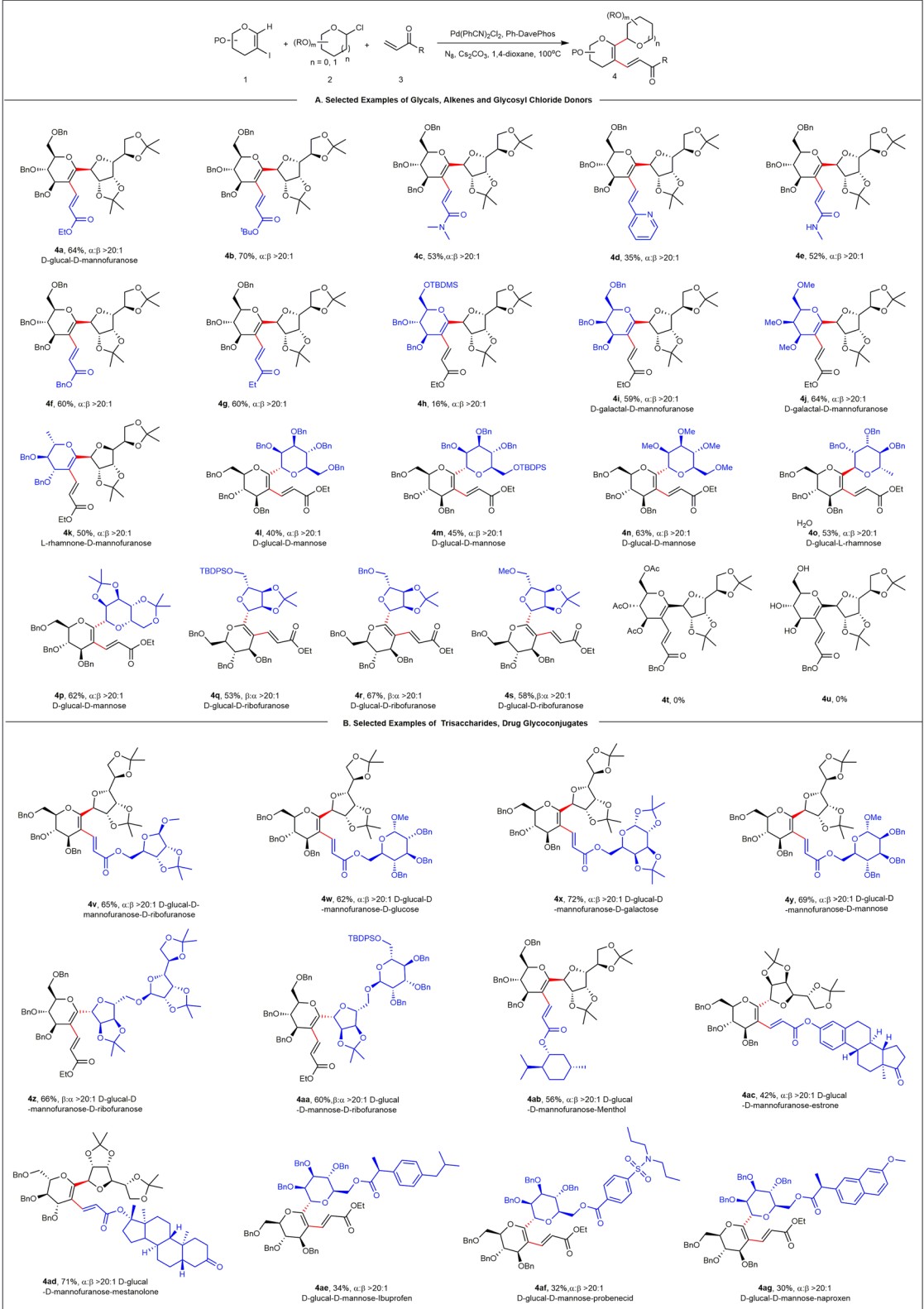

**Fig. 3 | Scope of C−H glycosylation of 2-iodoglycals via the alkenyl Catellani reaction[a,b].** [a]Reaction conditions: **1** (0.15 mmol), **2** (0.10 mmol), **3** (0.2 mmol), Pd(PhCN)$_2$Cl$_2$ (0.01 mmol), **L1** (0.02 mmol), **N$_8$** (0.2 mmol), Cs$_2$CO$_3$ (0.20 mmol), Ar, 100 °C, and 16 h. [b]Isolated yield.

including *tert*-butyl acrylate, acrylamide, vinyl ketone, and 2-vinylpyridine, well tolerated under the standard conditions, and the corresponding C-disaccharides **4a**–**4g** were obtained in 35−70% yield. Next, we examined the scope of 2-iodoglycals (derived from D-galactose and L-rhamnose), and the reactions of 2-iodoglycals **1c**, **1d**, and **1e** with **2a**

and **3a** resulted in the target products **4i**, **4j**, and **4k** with yields of 59%, 64%, and 50%, respectively. Conversely, the reaction cannot be applied to acetyl (Ac)-protected and unprotected sugars (**4t** and **4u**), possibly due to the coordination effect of ester groups on palladium[59] and the influence of *O*-glycosylation[60]. Furthermore, we examined the scope of

furanosyl and pyranosyl chloride donors. As shown in Fig. 3A, a series of methyl-, benzyl-, and TBDPS-protected α-mannosyl chlorides, α-rhamnosyl chloride, and β-ribofuranosyl chlorides with different substituents reacted to afford the corresponding products in good yields and with excellent stereoselectivities (**4l–4s**).

The power of our strategy for selective saccharide assembly of the reaction was demonstrated with the synthesis of trisaccharides, drug-conjugated sugar derivatives, and the late-stage modification of natural products (Fig. 3B). One pot efficient diastereoselective synthesis of trisaccharides was carried out using D-ribofuranose (**3h**), D-glucose (**3i**), D-galactose (**3j**), D-mannose (**3k**) as terminating reagents. Protected disaccharides (**4z** and **4aa**) were sequentially incorporated at the C1 position of 2-iodoglycals via the C-ribofuranose linkage with exclusive β-selectivity. Alkenes containing natural products, such as menthol, estrone, and metestosterone, were also found to be viable substrates, providing the desired products **4ab**, **4ac**, and **4ad** in 56%, 42%, and 71% yields, respectively. 1-Chloro-mannose-conjugated drug molecules, such as ibuprofen (nonsteroidal anti-inflammatory drug), probenecid (sulfonamide anti-gout drug), and naproxen (nonsteroidal anti-inflammatory drug), were synthesized, and were successfully C–H glycosylation to afford the corresponding products **4ae–4ag**.

## Synthetic utility

Thereafter, a large-scale attempt at cooperative catalysis was successfully conducted, and late-stage diversifications were carried out to prove the practical utility of our approach. Thus, a gram-scale reaction was carried out to afford **4a** in 46% yield. Likewise, cyanation was achieved using zinc cyanide to afford **5a** in 30% yield (Fig. 4A). Subsequently, it was demonstrated that Sonogashira-Hagihara termination was also applicable to C–H glycosylation. Desilylated carbohydrate product **7a** is a synthetic intermediate used in the preparation of various glycomimetics. For example, the Sonogashira coupling of **7a** with **7b** afforded C-disaccharide-phenylalanine conjugate **8a** in 72% yield. The C-disaccharide-triazole conjugate **9a** was

synthesized by a reaction using **7c** as a raw material in 80% yield. Furthermore, the tetrasaccharide **10a** was obtained by copper-catalyzed Glaser coupling of **7a** in 65% yield. Finally, under a copper catalyst, amide **11a** was efficiently synthesized by alkynyl conversion (Fig. 4B).

## Mechanistic investigation

Finally, DFT calculations were exploited to probe the mechanism of the palladium-catalyzed C−H glycosylation and determine the origin of the stereoselectivity. Trimethyl 2-iodoglucal (**12a**) and α-mannofuranose chloride (**2a**) were used as model substrates. Our calculated results showed that the energy barrier from **1h** to **D** requires 24.3 kcal/mol, which includes a series of processes such as the migratory insertion of norbornene and, subsequently, C1–H activation to form alkenyl-norbornyl-palladacycle (**D**) (Fig. 5A). Subsequently, **G-α** is received through **OA** (oxidative addition) and **RE** (reductive elimination) steps. The DFT calculations revealed that the **OA** proceeds via a concerted three-membered cyclic transition state (**4-TS**), which directly leads to the form of the D-mannofuranosyl Pd(IV) intermediate (**F-α**) without the formation of oxocarbenium[26,56] (Supplementary Figs. 5 and 10). Subsequently, the **RE** process gives **G-α** through the three-membered cyclic transition state (**5-TS**). In conclusion, the **OA** and **RE** transition states in the α-selective pathway have relatively low barriers, whereas the corresponding β-mannofurglycosylation pathway of the β-mannofuranose chloride (**2a′**) is clearly unfavorable. In **F-α**, phosphine and α-mannofuranose are on the opposite side of glucal, which leads to small steric hindrances. Nevertheless, there could be strong repulsive interactions between different groups in **F-β**. Therefore, **F-β** is located 28.8 kcal/mol higher in energy than **F-α**, and the energy of **4-TS′** is much higher than **4-TS** (Fig. 5B).

## Discussion

In conclusion, we have developed the palladium-catalyzed nondirected late-stage C($sp^2$)–H glycosylation of 2-iodoglycals, resulting in the

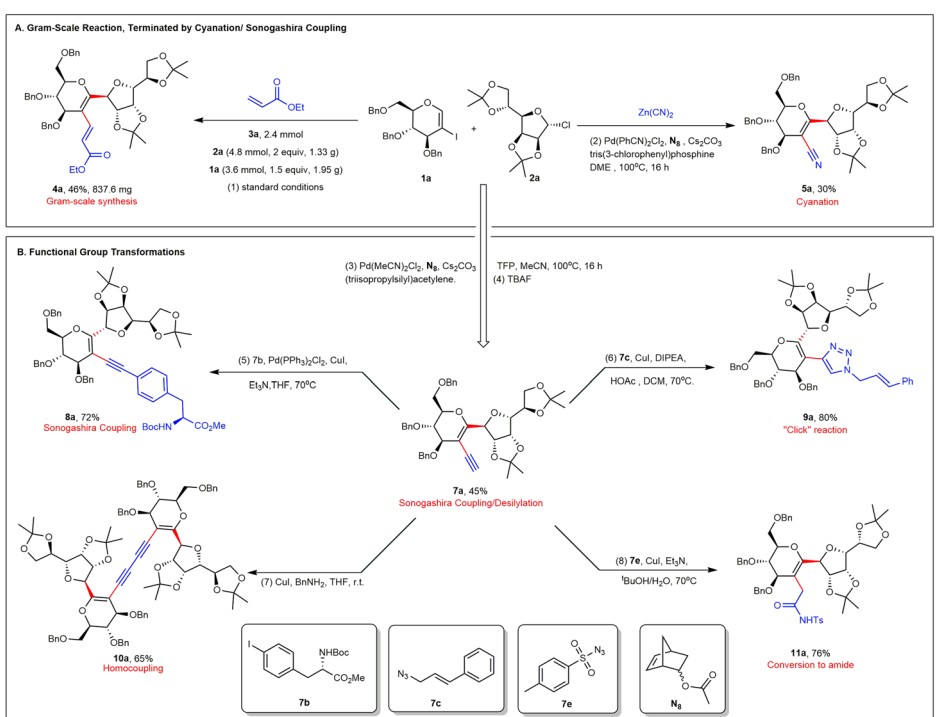

**Fig. 4 | Synthetic applicability. A** Gram-scale reaction terminated by cyanation/ Sonogashira coupling. **B** Functional group transformations. (1) Gram-scale reaction to synthesize C-disaccharide under standard conditions (2–4). Cyanation, Sonogashira coupling, and removing triisopropylsilyl. (5–8) Diverse transformations

using C-disaccharides. TFP tri(2-furyl)phosphine, TBAF *tetra*-n-butylammonium fluoride, DIPEA *N,N*-diisopropylethylamin. See the Supplementary Information for experimental details.

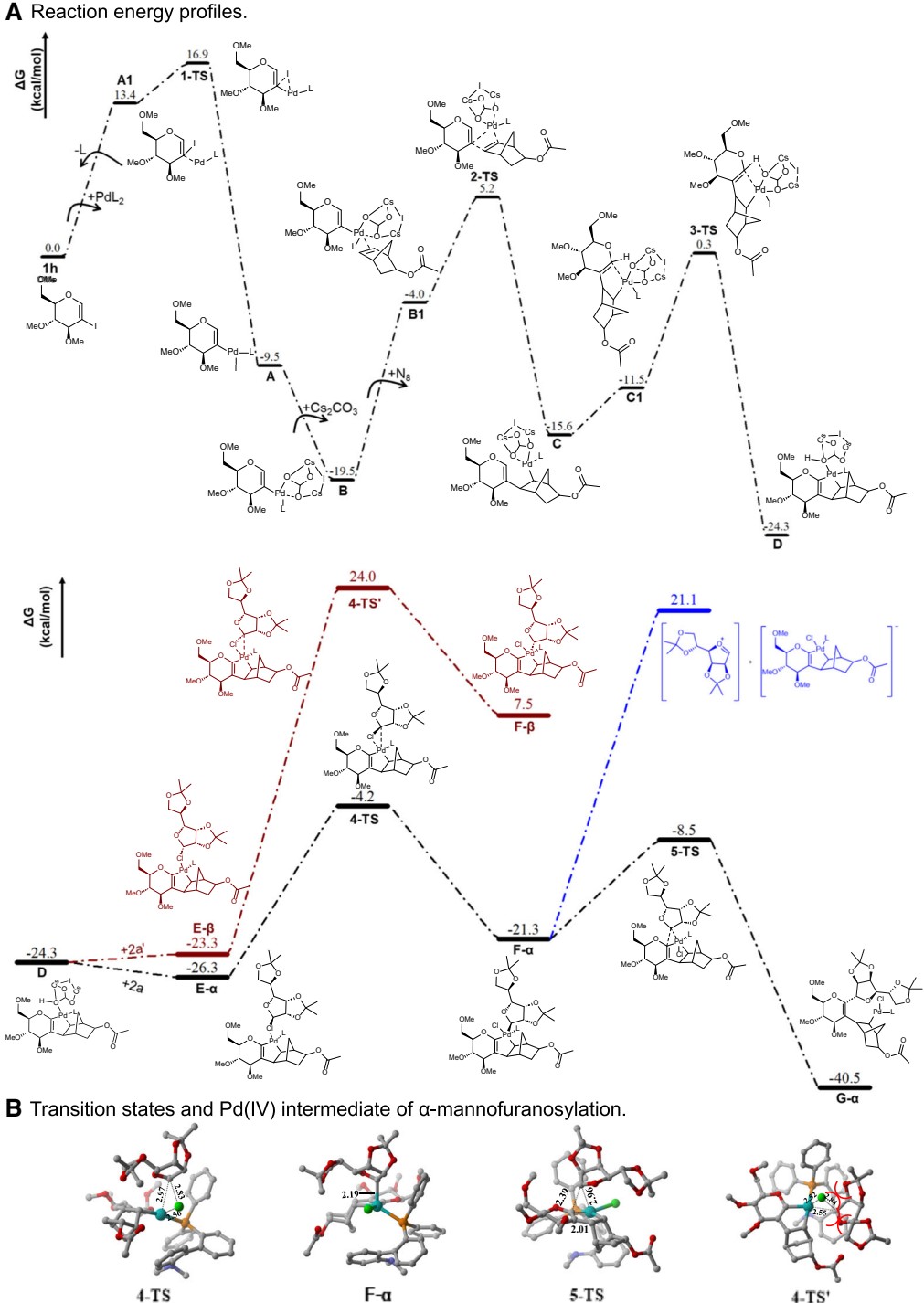

**A** Reaction energy profiles.

**B** Transition states and Pd(IV) intermediate of α-mannofuranosylation.

**Fig. 5 | DFT calculations. A** Computational studies of the Pd-catalyzed C−H mannofuranosylation of 2-iodoglycal **12a**. **B** Transition states and Pd(IV) intermediate of α-mannofuranosylation. Computed free energy surface. Bond distances are in Å. A detailed introduction of computational methods can be found in Supplementary Information.

modular assembly of unnatural C-disaccharides, C-trisaccharides, and drug glycoconjugates. This strategy does not require multiple steps during synthesis to prepare the complex precursor and does not need directing groups, along with their introduction and removal. The strategy of cooperative catalysis was highly diastereoselective, widely applicable, and operationally simple. Based on density functional theory (DFT) calculations, we suggest that the stereo-selective C−H glycosylation undergoes oxidative addition/reduction elimination. We anticipate that our strategy may aid the future design of new synthesis strategies for bio-relevant C-oligosaccharides in carbohydrate chemistry.

## Methods

### Experimental procedure: stereoselective assembly of C-oligosaccharides

In a dried 10 ml tube equipped with a stirring bar, 1 (0.15 mmol, 1.5 equiv), 2 (0.2 mmol, 2.0 equiv), Pd(PhCN)$_2$Cl$_2$ (0.01 mmol, 0.1 equiv), Ph-Davephos (0.02 mmol, 0.2 equiv), and Cs$_2$CO$_3$ (0.2 mmol, 2.0 equiv) were added. The tube charged with argon more than three times. 3 (0.1 mmol, 1.0 equiv), N$_8$ (0.2 mmol, 2.0 equiv), and 1,4-dioxane (1.0 ml) were injected into the tube via microsyringe and plastic syringes, respectively. The resulting suspension was placed in an oil

bath that had been preheated to 100 °C for 16 h, and then the mixture was cooled to r.t. The reaction mixture was filtered and concentrated in vacuo. The residue was purified with a chromatography column on silica gel to give 4 (petroleum ether/EtOAc).

## Data availability
The data generated or analyzed during this study are included in this article and the supplementary information. The Cartesian coordinates are available from the Source Data. Details about materials and methods, experimental procedures, characterization data, computational details, and NMR spectra are available in the Supplementary Information. All data are available from the corresponding author upon request. Source data are provided with this paper.

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

## Acknowledgements

We are grateful to the National Natural Science Foundation of China (22171114, Y.-M.L. and 22371097, Y.-M. L.).

## Author contributions

X.-Y.L. and Y.-M.L. conceived the projects. Y.-N.D, M.-Z.X. and Y.-C.H. performed the experiments under the supervision of Y.-M.L. Y.-M.L. and L.A. wrote the manuscript with the feedback of all other authors. Theoretical calculations were carried out by X.-T.K.

## Competing interests

The authors declare no competing interests.
