## [Peer Review File · Nature Communications]

Stereoselective Assembly of C-oligosaccharides via Modular Difunctionalization of GlycalsREVIEWER COMMENTS

Reviewer #1 (Remarks to the Author):

C-oligosaccharides are found in many bioactivity relevant molecules including natural products and drugs. However, the efficient access to them has been challenging the existing synthetic methods. To solve this problem, the authors established a novel protocol to prepare C-oligosaccharide derivatives, which is featured by palladium-catalyzed three-component Catellani-type reaction between 2-iodoglycals, glycosylchlorides, and various terminating reagents. The method exhibited excellent stereoselectivity and enjoyed broad substrate scope. Moreover, the reaction mechanism was also investigated by DFT calculation to explain the excellent stereoselectivity of the reactions. In the Supporting Information part, all new compounds have been thoroughly characterized. Thus, the manuscript could be considered to publish in Nat. Commun. provided that the following problems have been properly solved.

- 1) The authors noticed that acetylated and unprotected glycols were not viable substrates for the established 2-iodoglycal-involved Catellani reactions, a reasonable explanation should be provided in the main text;
- 2) Some products were not pure enough as evidenced from the corresponding spectra provided in the Supporting Information part, particularly for compounds 4d, 4g, 4h, 4i, 4l, 4m, and 4o, which made me suspect the yields as well as stereoselectivity reported in the manuscript;
- 3) The language should be further improved to make the manuscript more readable. For instances, the sentence in line 15 "Density functional theory", "thuas" in line 24, "C-at two anomeric carbons" in line 48, "ligands and bases.ethyl crylate." in line 87; "including ethyl tert-butyl" in line 117, the sentence "In addition to" in line 123, and the sentence "High-efficiency diastereoselective" in line 140;
- 4) In the reference part, the format should be modified. For most of references, the journal names were provided in abbreviation form, while the journal names were presented in the full form in ref.38, 39, 42, 45, 51, 53. Most importantly, the journal name for ref.57 was missing.

Reviewer #2 (Remarks to the Author):

The authors described palladium-catalyzed nondirected C(sp²)-H glycosylation of 2-iodoglycals based on Catellani-type reaction. The reactions showed high diastereoselectivity and broad substrate scope and readily provided a variety of unique C-disaccharides with branched alkenyl chains. The authors proposed a plausible mechanism by applying DFT calculations. Because the manuscript demonstrates the high versatility of the method to access the unprecedented C-disaccharide structures and suggests an interesting mechanism for the stereoretention of the anomeric position, it potentially interests the readers of Nature Communications. However, the following points need to be addressed before the publication.

1. In the mechanistic study section, Gibbs free energies were computed at the PBE0-D3(BJ)/def2-TZVP level. However, due to the presence of many electron-negative oxygen and chlorine atoms, diffusion functions should be incorporated in the basis set to improve the accuracy (e.g., def2-TZVPD).
2. In the mechanistic study section, the authors suggested a concerted oxidative addition to

a glycosyl chloride substrate. Other potential pathways were ruled out based on results from the IRC calculation and the relaxed scan of the C–Cl bond. However, the IRC calculation confirms the sufficiency of a specific transition state, and the relaxed scan experiment may lead to a local minimum. Because these experiments do not ensure that a specific transition state is the most stable transition state, other reaction pathways, such as the SN2 and SN1 pathways, cannot be excluded from the present computational study only. Although the SN2 pathway can be ruled out from the observed anomeric stereoretention of glycosyl chlorides, the SN1 pathway, proposed by the authors in their former paper (J. Org. Chem. 2020, 85, 11280.), remains plausible. To address this point, the dissociation energy of the corresponding glycosyl cation from the Pd(IV) species should be computed as previously calculated by the authors (CCS Chem. 2020, 2, 1729).

3. In Figure S1, valency “II” is assigned to the palladium metal in transition states. However, this is not accurate according to the chemical definition of valency, and these notations should be omitted from all transition states.

4. On page 1, the 7th line of the main text, there is a mention of “alkenylation¹⁹⁻²⁰”. However, since reference 20 primarily discussed alkylation, it would be more accurate to state “alkenylation,¹⁹ alkylation,²⁰”

5. In Supplementary Information, there are many errors that require corrections.

a. The TMS peak of the ¹H NMR spectrum of compound 4w is designated as 0.5 ppm. Consequently, chemical shifts in this compound are shifted by +0.5 ppm.

b. The ¹H NMR peak list of compound 4ae lacks 3 protons. Specifically, only 75 protons are shown, while it should have a total of 78 protons.

c. The molecular formulas for HRMS of compounds 4e, 5a, and 8a lack one nitrogen. For example, the correct formula for compound 4e is C₄₃H₅₁NNaO₁₀, not C₄₃H₅₁NaO₁₀.

d. Although the authors state that “chemical shifts are recorded in ppm relative to the residual solvent signal (TMS δ = 0 for ¹H NMR),” the TMS peaks are absent in some ¹H NMR spectra (i.e., compound 2n). Additionally, on page S1, there are two sentences about NMR reference. (i.e., “Chemical shifts ... signal (TMS δ = 0 for ¹H NMR and CDCl₃ δ = 77.0 for ¹³C NMR).” and “Chemical shifts ... with TMS (tetramethylsilane) as the internal reference standard”.) Given the absence of TMS peaks in ¹³C NMR charts, the latter sentence should be deleted.

6. The characterization data provided in the Supplementary Information is not adequate for the structural determination.

a. Although the NOESY correlations are essential for confirming the depicted C-glycosidic linkages, only the NOESY spectra of compounds 4j, 4p, and 4s are included. The glycosidic stereochemistries of the other compounds should be determined based on the NOESY correlations or the J-values of proton peaks.

b. Optical rotations and melting points should be reported for all chiral compounds and solid compounds.

c. Some new compounds lack HRMS data (e.g., compounds 2k, 2l, 2h1, 2m, 2n, 2o, 3i, 3k, 3l).

d. ¹H NMR spectra should consistently be listed with three significant digits. Compounds 4m, 4v, and 4ae have only two significant digits.

6. This manuscript and Supplementary Information have many errors and typos. Some of them are listed below.

a. Page 1, the 6th line from the bottom

“chen’s group” -> “Chen’s group”

b. Page 2, Fig. 1 (both a figure and a legend)

“Isoquinolin” -> “Isoquinoline”

c. Page 2, the 9th and 10th lines from the bottom

“Stille couplings (Fig. 2b) and the radical-radical homocouplings (Fig.2c) of glycols” -> “Stille couplings (Fig. 2b) of glycols and the radical-radical homocouplings (Fig.2c) of glycosides”

(i.e., Substrates in Fig.2c are not glycols but glycosides)

d. Page 3, Fig. 2c, the chemical structure of acyl telluride

“PhTeCO” -> “COTePh”

e. Page 3, Fig. 2d

The glycol of the Pd(IV) intermediate is missing the letter “n” near the parenthesis.

f. Page 4, the 7th line of Results and discussion section

“ethyl acrylate.” is unnecessary.

g. Page 4, the 14th line of Results and discussion section,

“entries 11-13” -> “entries 8, 11-13”

h. Page 4, the 4th line from the bottom

“smNBEs” -> “smNBE”

i. Page 5, Table 1, the chemical structure of N4.

The ester moiety should be oriented in the same plane as olefins.

j. Page 5, the 2nd line of Substrate scope section

“glycoacyl” -> “glycosyl”

k. Page 7, line 4

“acetyl (OAc)” -> “acetyl (Ac)”

l. Page 8, the 1st line of Synthetic utility section

“Thereafte” -> “Thereafter”

m. Page 8, the 4th line of Synthetic utility section

“(triisopropylsilyl)acetylene” -> “zinc cyanide”

n. page 10, reference 8

“(2022)” -> “(2023)”

o. Page 11, reference 57

The citation lacks the journal name “Angew. Chem. Int. Ed.”.

p. Page S1, the 6th line of Instrumentation section.

The “3” of “CDCl3” should be a subscript.

Reviewer #3 (Remarks to the Author):

The authors present a new method to synthesize C-oligosaccharides through Catellani-type three-component coupling involving 2-iodoglycols and glycosyl chlorides. The developed method is remarkable because it stereoselectively provides various C-oligosaccharides in a single operation. Moreover, several C-oligosaccharide conjugate molecules were synthesized using this method. Therefore, this work is worthy to publish, but the manuscript and Supporting Information should be thoroughly revised and resubmitted because scholarly descriptions are inappropriate. Additional comments are provided below.

(1) L24, “thusas”: Typo?

(2) L38-41, “C-oligosaccharides are pharmacologically relevant compounds ~ , rendering them preferred chemotherapy drugs, such as maitotoxin, ~”: Maitotoxin is a poisonous compound rather than a drug. The descriptions on maitotoxin should be removed from the main text and Fig. 2. Ref 29 should also be deleted because it describes the total synthesis of hikizimycin.

- (3) L45: Ref 35 is inappropriate because a natural product synthesis rather than a C-oligosaccharide synthesis was reported.
- (4) L48, "C-at two anomeric carbons": This part is unclear.
- (5) L50, "polarity reversal at the anomeric": Typo?
- (6) L52-53, "and have limited substrate scope (Fig. 2a).": Fig. 2a shows some bioactive compounds rather than the reaction scope.
- (7) L81-82, "As shown in Table 1, ~conditions for the envisioned palladium/norbornene (NBE)-enabled ~ via cooperative catalysis ~": "palladium/norbornene (NBE)-enabled" and "via cooperative catalysis" are almost the same. Delete the latter.
- (8) L87, "bases.ethyl acrylate": Typo
- (9) L95: Is "Ph-DavePhos" L1? Indicate clearly.
- (10) L100: What is "ANP"?
- (11) L103: The reason why N8 is optimal should be provided (see below).
- (12) L111, "2-iodioses": Typo?
- (13) L117, "ethyl tert-butyl": typo?
- (14) L125, (4h-4k)": typo? "4i-4k" is correct?
- (15) Fig. 4A: "Gram-scale synthesis" is incorrect because the amount of starting 2-iodoglucal is less than 1 g and 471.1 mg of 4a was obtained. The label "Synthetic amide" is also strange.
- (16) L141-142, "D-ribofuranose (4v), D-glucose (4w), D-galactose (4x), D-mannose (4y) as terminating reagents,": This description is strange because 4v-4y are the products rather than terminating agents. "D-ribofuranose-, D-glucose-, D-galactose-, and D-mannose-derived acrylates" are the terminating agents.
- (17) L153, "Thereafte", "succsfully": Typos?
- (18) L161, (Fig.4A): typo? "Fig. 4B" is correct?
- (19) L173-174, "the total energy barrier is -25.7 kcal mol⁻¹": This description is very strange. What does an energy barrier with a negative value mean?
- (20) L179-180: Figure S2 shows the IRC result for OA through 4-TS rather than that for RE through 5-TS.
- (21) DFT calculations using NBE should give incorrect estimations of energy surfaces because NBE is much less efficient than N8, which is experimentally used. Therefore, DFT calculations should be conducted using N8. Compare results obtained from DFT calculations using other NBEs to discuss why N8 is optimal.
- (22) L216: "n/a" => "62"
- (23) SI: Are starting materials known compounds? If so, references, which report full characterization data, should be cited for each compound. For new compounds, MS data should be provided.
- (24) The deviation of the MS data for 4m from calcd value is over 0.0030.
- (25) The deviation of the MS data for 4ac from calcd value is over 0.0030.
- (26) The deviation of the MS data for 4ad from calcd value is over 0.0030.
- (27) The deviation of the MS data for 8a from calcd value is over 0.0030.

REVIEWER COMMENTS

Reviewer #1 (Remarks to the Author):

C-oligosaccharides are found in many bioactivity relevant molecules including natural products and drugs. However, the efficient access to them has been challenging the existing synthetic methods. To solve this problem, the authors established a novel protocol to prepare C-oligosaccharide derivatives, which is featured by palladium-catalyzed three-component Catellani-type reaction between 2-iodoglycals, glycosyl chlorides, and various terminating reagents. The method exhibited excellent stereoselectivity and enjoyed broad substrate scope. Moreover, the reaction mechanism was also investigated by DFT calculation to explain the excellent stereoselectivity of the reactions. In the Supporting Information part, all new compounds have been thoroughly characterized. **Thus, the manuscript could be considered to publish in Nat. Commun.** provided that the following problems have been properly solved.

Reply: We thank the reviewer for the positive recommendation. Much appreciated!

1) The authors noticed that acetylated and unprotected glycols were not viable substrates for the established 2-iodoglycal-involved Catellani reactions, a reasonable explanation should be provided in the main text;

Reply: Thank you very much for your comment. For unprotected glycols, we speculate that the possible reason is that the ester groups have certain coordination effects on palladium (*ACS Catal.* **2018**, *8*, 7781–7786). On the other hand, we detected a large number of by-products of O-glycosylation in the **4u** system (*CCS Chem.* **2020**, *2*, 1821–1829. 10.31635/ccschem.020.202000445). We speculate that the possible reason is that the reaction is more inclined towards O-glycosylation rather than the Catellani reaction process. Possible reasons have been added to manuscript. (Manuscript page 7, line 2-3)

2) Some products were not pure enough as evidenced from the corresponding spectra provided in the Supporting Information part, particularly for compounds **4d**, **4g**, **4h**, **4i**, **4l**, **4m**, and **4o**, which made me suspect the yields as well as stereoselectivity reported in the manuscript;

Reply: We apologize for any confusion caused by the impurity of the target products. Compounds **4d**, **4g**, **4h**, **4i**, **4l**, **4m**, and **4o** were carefully tested again and purified. We found a small decrease in yield, but no change in stereoselectivity. (Manuscript page 7, Fig. 3.and Supporting Information NMR Spectroscopic Data).

3) The language should be further improved to make the manuscript more readable. For instances, the sentence in line 15 “Density functional theory”, “thuas” in line 24, “C-at two anomeric carbons” in line 48, “ligands and bases.ethyl crylate.” in line 87; “including ethyl tert-butyl” in line 117, the sentence “In addition to” in line 123, and the sentence “High-efficiency diastereoselective” in line 140;

Reply: Thank you for your professional suggestions. We have further improved the language to make the manuscript more readable.

The sentence “Density functional theory (DFT) calculations studies are supportive of a concerted oxidative addition mechanism alkenyl-norbornadiene-palladacycle (**ANP**) intermediate with an α -mannofuranose chloride and the high stereoselectivity of glycosylation was due to steric hindrance.” was changed to “The results of density functional theory (DFT) calculations support oxidative addition mechanism of alkenyl-norbornadiene-palladacycle (**ANP**) intermediate with α -mannofuranose chloride and the high stereoselectivity of glycosylation was due to steric hindrance.” (Manuscript page 1. Abstract, line 11-14).

The sentence “monofunctionalization **thuas** far typically involved Heck.” was changed to “monofunctionalization typically involves palladium-catalyzed Heck.” (Manuscript page 1, line 6-7).

The sentence “In addition, methods that directly provide access to **C-at two anomeric carbons** are rare.” was changed to “In addition, the methods for assembling C-disaccharides that involve directly connecting the anomeric carbon and anomeric carbon are rare.” (Manuscript page 2, line 11-12).

The sentence “Compound **4a'** was obtained as a byproduct via direct Mizoroki-Heck coupling between 2-iodoglycal, which can be inhibited by screening **ligands and bases.ethyl acrylate.**” was corrected to “Compound **4a'** was obtained as byproduct *via* direct Mizoroki-Heck coupling, which can be inhibited by screening ligands and NBEs.” (Manuscript page 4, line 13-14).

The sentence “Various olefin terminating reagents, including **ethyl tert-butyl**, amide, ketone, and pyridine were well tolerated” was corrected to “Various olefin terminating reagents, including **tert-butyl acrylate**, amide, ketone, and pyridine were well tolerated.” (Manuscript Page 5, line 8 of Substrate scope.)

The sentence “In addition to D-2-iodoglucal, the reactions of glycals **1i–1k**, derived from D-galactose and L-rhamnose, with glycosyl chloride donor **2a** and acrylate **3a** were also compatible under the reaction conditions (**4h–4k**.)” was changed to “Next, we examined the scope of 2-iodoglycals (derived from D-galactose and L-rhamnose), and the reactions of 2-iodoglycals **1c**, **1d**, and **1e** with **2a** and **3a** were gave target products **4i**, **4j**, and **4k** with yields of 59%, 64%, and 50%, respectively.” (Manuscript page 6, line 1-3).

The sentence “High-efficiency diastereoselective one-pot synthesis of trisaccharides was conducted using D-ribofuranose (**4v**), D-glucose (**4w**), D-galactose (**4x**), D-mannose (**4y**) as terminating reagents with three-component reactions of 2-iodoglycal **1a** and glycosyl chloride donor **2a**.” was corrected to “One pot efficient diastereoselective synthesis of trisaccharides were carried out using D-ribofuranose (**3h**), D-glucose (**3i**), D-galactose (**3j**), D-mannose (**3k**) as terminating reagents.” (Manuscript page 7, line 10-12).

In addition, we carefully checked the entire manuscript and corrected some incorrect expressions highlighted in the manuscript.

4) In the reference part, the format should be modified. For most of references, the journal names were provided in abbreviation form, while the journal names were presented in the full form in ref.38, 39, 42, 45, 51, 53. Most importantly, the journal name for ref.57 was missing.

Reply: Thanks for your corrections. We have corrected above typos in the references. In addition, we have referred to the correct reference format and carefully revised all reference formats. Please see the references.

Reviewer #2 (Remarks to the Author):

The authors described palladium-catalyzed nondirected C(sp²)-H glycosylation of 2-iodoglycals based on Catellani-type reaction. The reactions showed high diastereoselectivity and broad substrate scope and readily provided a variety of unique C-disaccharides with branched alkenyl chains. The authors proposed a plausible mechanism by applying DFT calculations. Because the manuscript demonstrates the high versatility of the method to access the unprecedented C-disaccharide structures and suggests an interesting mechanism for the stereoretention of the anomeric position, **it potentially interests the readers of Nature Communications**. However, the following points need to be addressed before the publication.

Reply: We thank the reviewer for the positive recommendation. Much appreciated!

1. In the mechanistic study section, Gibbs free energies were computed at the PBE0-D3(BJ)/def2-TZVP level. However, due to the presence of many electron-negative oxygen and chlorine atoms, diffusion

functions should be incorporated in the basis set to improve the accuracy (e.g., def2-TZVPD).

Reply: Thanks very much for the comment. The def2-TZVPD basis set was used for the single point energy corrections. In addition, to fully reveal the energy changes in homogeneous reactions, the solvation free energies were considered in our revised manuscript. The detailed explanations could be found in the Computational Details in SI.

2. In the mechanistic study section, the authors suggested a concerted oxidative addition to a glycosyl chloride substrate. Other potential pathways were ruled out based on results from the IRC calculation and the relaxed scan of the C–Cl bond. However, the IRC calculation confirms the sufficiency of a specific transition state, and the relaxed scan experiment may lead to a local minimum. Because these experiments do not ensure that a specific transition state is the most stable transition state, other reaction pathways, such as the S_N2 and S_N1 pathways, cannot be excluded from the present computational study only. Although the S_N2 pathway can be ruled out from the observed anomeric stereoretention of glycosyl chlorides, the S_N1 pathway, proposed by the authors in their former paper (*J. Org. Chem.* **2020**, 85, 11280.), remains plausible. To address this point, the dissociation energy of the corresponding glycosylation from the Pd(IV) species should be computed as previously calculated by the authors (*CCS Chem.* **2020**, 2, 1729).

Reply: Thanks very much for your professional comment. We calculated the dissociation energy based on Chen's work (*CCS Chem.* **2020**, 2, 1729). The Pd–C1 bond of the Pd(IV) species (F- α) dissociates to form the oxocarbenium ion with absorbing 42.4 kcal/mol, which hints that the Pd(IV) intermediate is configurationally stable. (see Supporting Information for details).

3. In Figure S1, valency “II” is assigned to the palladium metal in transition states. However, this is not accurate according to the chemical definition of valency, and these notations should be omitted from all transition states.

Reply: Thank you for your professional suggestion. We have removed valency “II”. (Supporting information, Figure S1)

4. On page 1, the 7th line of the main text, there is a mention of “alkenylation19-20”. However, since reference 20 primarily discussed alkylation, it would be more accurate to state “alkenylation,19 alkylation, 20”

Reply: Thank you for your professional suggestion. We have corrected the citation in manuscript. (Manuscript page 1, line 8).

5. In Supplementary Information, there are many errors that require corrections.

Reply: Thanks for your corrections and we are sorry for our negligence. We have corrected some typos in supporting information.

a. The TMS peak of the ^1H NMR spectrum of compound **4w** is designated as 0.5 ppm. Consequently, chemical shifts in this compound are shifted by +0.5 ppm.

Reply: Thanks for your correction. The TMS peak ^1H NMR spectrum and corresponding data of the compound **4w** was corrected in supporting information.

b. The ^1H NMR peak list of compound **4ae** lacks 3 protons. Specifically, only 75 protons are shown, while it should have a total of 78 protons.

Reply: Thanks for your correction. The peak ^1H NMR spectrum and corresponding data of the compound **4ae** was corrected in supporting information.

c. The molecular formulas for HRMS of compounds **4e**, **5a**, and **8a** lack one nitrogen. For example, the correct formula for compound **4e** is $\text{C}_{43}\text{H}_{51}\text{N}\text{NaO}_{10}$, not $\text{C}_{43}\text{H}_{51}\text{NaO}_{10}$.

Reply: Thanks for your correction. We have corrected above typos in supporting information.

4e: " $\text{C}_{43}\text{H}_{51}\text{NaO}_{10}$ " was corrected to " $\text{C}_{43}\text{H}_{51}\text{NNaO}_{10}$ "

5a: " $\text{C}_{40}\text{H}_{45}\text{NaO}_9$ " was corrected to " $\text{C}_{40}\text{H}_{45}\text{NNaO}_9$ "

8a: " $\text{C}_{56}\text{H}_{65}\text{NaO}_{13}$ " was corrected to " $\text{C}_{56}\text{H}_{65}\text{NO}_{13}$ "

d. Although the authors state that "chemical shifts are recorded in ppm relative to the residual solvent signal (TMS $\delta = 0$ for ^1H NMR)," the TMS peaks are absent in some ^1H NMR spectra (i.e., compound **2n**). Additionally, on page S1, there are two sentences about NMR reference. (i.e., "Chemical shifts ... signal (TMS $\delta = 0$ for ^1H NMR and CDCl_3 $\delta = 77.0$ for ^{13}C NMR)." and "Chemical shifts ... with TMS (tetramethylsilane) as the internal reference standard".) Given the absence of TMS peaks in ^{13}C NMR charts, the latter sentence should be deleted.

Reply: Thanks for your corrections. We have corrected the TMS peak of ^1H NMR spectra in compound **2n**. In addition, we also examined other spectras and corrected the TMS peak of ^1H NMR spectra in compound **2l**, **3n**.

The sentence "Chemical shifts (ppm) were recorded with TMS (tetramethylsilane) as the internal reference standard." was deleted.

6. The characterization data provided in the Supplementary Information is not adequate for the structural determination.

a. Although the NOESY correlations are essential for confirming the depicted C-glycosidic linkages, only the NOESY spectra of compounds **4j**, **4p**, and **4s** are included. The glycosidic stereochemistries of the other compounds should be determined based on the NOESY correlations or the J-values of proton peaks.

Reply: Thank you for your professional suggestion. We added the NOESY spectra of compounds **4d**, **4g**, **4i**, **4o**, **4l**, **4r**, **11a** in the supporting information. For other carbohydrate compounds of the same type, the stereoscopic configuration is determined by the chemical shift and J-values of anomeric H and marked in the characterization data.

b. Optical rotations and melting points should be reported for all chiral compounds and solid compounds.

Reply: Thank you for your professional suggestion. The optical rotation data of all products and melting points (**4e**, **4p**, **4x**, **4ad**, **8a**, **10a**) were supplemented and added in supporting information.

4a, $[\alpha]_{\text{D}}^{24}=60.0$ (c=5 mg/ml, CH_2Cl_2)

4b, $[\alpha]_{\text{D}}^{24}=78.0$ (c=5 mg/ml, CH_2Cl_2)

4c, $[\alpha]_{\text{D}}^{24}=104.0$ (c=5 mg/ml, CH_2Cl_2)

4d, $[\alpha]_D^{25}=194.0$ (c=5 mg/ml, CH₂Cl₂)
4e, $[\alpha]_D^{25}=124.0$ (c=5 mg/ml, CH₂Cl₂), melting point: 73-75°C.
4f, $[\alpha]_D^{25}=74.0$ (c=5 mg/ml, CH₂Cl₂)
4g, $[\alpha]_D^{25}=152.0$ (c=5 mg/ml, CH₂Cl₂)
4h, $[\alpha]_D^{25}=44.0$ (c=5 mg/ml, CH₂Cl₂)
4i, $[\alpha]_D^{25}=74.0$ (c=5 mg/ml, CH₂Cl₂)
4j, $[\alpha]_D^{25}=102.0$ (c=5 mg/ml, CH₂Cl₂)
4k, $[\alpha]_D^{25}=70.0$ (c=5 mg/ml, CH₂Cl₂)
4l, $[\alpha]_D^{25}=39.0$ (c=10 mg/ml, CH₂Cl₂)
4m, $[\alpha]_D^{25}=40.0$ (c=5 mg/ml, CH₂Cl₂)
4n, $[\alpha]_D^{25}=80.0$ (c=5 mg/ml, CH₂Cl₂)
4o, $[\alpha]_D^{25}=-10.0$ (c=1 mg/ml, CH₂Cl₂)
4p, $[\alpha]_D^{25}=100.0$ (c=2 mg/ml, CH₂Cl₂); melting point: 58-60°C.
4q, $[\alpha]_D^{25}=-22.0$ (c=5 mg/ml, CH₂Cl₂)
4r, $[\alpha]_D^{25}=-40.0$ (c=5 mg/ml, CH₂Cl₂)
4s, $[\alpha]_D^{25}=-24.0$ (c=5 mg/ml, CH₂Cl₂)
4v, $[\alpha]_D^{25}=64.0$ (c=5 mg/ml, CH₂Cl₂)
4w, $[\alpha]_D^{25}=94.0$ (c=5 mg/ml, CH₂Cl₂)
4x, $[\alpha]_D^{25}=49.0$ (c=10 mg/ml, CH₂Cl₂); melting point: 80-82°C.
4y, $[\alpha]_D^{25}=67.0$ (c=5 mg/ml, CH₂Cl₂)
4z, $[\alpha]_D^{25}=-12.0$ (c=5 mg/ml, CH₂Cl₂)
4aa, $[\alpha]_D^{25}=-11.0$ (c=10 mg/ml, CH₂Cl₂)
4ab, $[\alpha]_D^{25}=71.0$ (c=10 mg/ml, CH₂Cl₂)
4ac, $[\alpha]_D^{25}=110.0$ (c=10 mg/ml, CH₂Cl₂)
4ad, $[\alpha]_D^{25}=92.0$ (c=10 mg/ml, CH₂Cl₂); melting point: 69-71°C.
4ae, $[\alpha]_D^{25}=26.0$ (c=5 mg/ml, CH₂Cl₂)
4af, $[\alpha]_D^{25}=58.0$ (c=5 mg/ml, CH₂Cl₂)
4ag, $[\alpha]_D^{25}=38.0$ (c=5 mg/ml, CH₂Cl₂)
5a, $[\alpha]_D^{25}=38.0$ (c=5 mg/ml, CH₂Cl₂)
6a, $[\alpha]_D^{25}=47.0$ (c=5 mg/ml, CH₂Cl₂)
7a, $[\alpha]_D^{25}=54.0$ (c=5 mg/ml, CH₂Cl₂)
8a, $[\alpha]_D^{25}=43.0$ (c=5 mg/ml, CH₂Cl₂); melting point: 62-64°C.
9a, $[\alpha]_D^{25}=26.0$ (c=10 mg/ml, CH₂Cl₂)
10a, $[\alpha]_D^{25}=61.0$ (c=5 mg/ml, CH₂Cl₂) ; melting point: 74-76°C.
11a, $[\alpha]_D^{25}=-8.0$ (c=10 mg/ml, CH₂Cl₂)

c. Some new compounds lack HRMS data (e.g., compounds **2k**, **2l**, **2h**, **2m**, **2n**, **2o**, **3i**, **3k**, **3l**).

Reply: Thank you for your professional suggestion. The HRMS data for the new compounds (**1b**, **2k**, **2l**, **2h**, **2h-1**, **2m**, **2n**, **2o**, **3h**, **3i**, **3k**, **3l**) have been added to the supporting information.

1b: HRMS (ESI) m/z: [M + Na]⁺ Calcd for C₂₆H₃₅IO₄SiNa 589.1242 found 589.1247.

2k: HRMS (ESI) m/z: [M + Na]⁺ Calcd for C₂₀H₃₁ClO₉Na 473.1549; Found 473.1538.

2l: HRMS (ESI) m/z: [M + Na]⁺ Calcd for C₅₁H₅₉ClO₉SiNa 901.3509; Found 901.3549.

2h: HRMS (ESI) m/z: [M + Na]⁺ Calcd for C₄₃H₄₇ClO₅SiNa 729.2773; Found 729.2750.

2h-1: HRMS (ESI) m/z: [M + Na]⁺ Calcd for C₂₇H₂₉ClO₅Na 491.1596; Found 491.1577

2m: HRMS (ESI) m/z: [M + Na]⁺ Calcd for C₄₀H₄₅ClO₆Na 679.2797; Found 679.2782.

2n; HRMS (ESI) m/z: [M + Na]⁺ Calcd for C₄₀H₄₅ClO₆Na 758.2525; Found. 758.2517.

2o: HRMS (ESI) m/z: [M + Na]⁺ Calcd for C₄₁H₄₁ClO₇Na 703.2433; Found 703.2409.

3h: HRMS (ESI) m/z: [M + Na]⁺ Calcd for C₁₂H₁₈O₆Na 281.0996 Found 281.0995.

3i: HRMS (ESI) m/z: [M + Na]⁺ Calcd for C₃₁H₃₄O₇Na 541.2197; Found 541.2169.

3k: HRMS (ESI) m/z: [M + Na]⁺ Calcd for C₃₁H₃₄O₇Na 541.2197; Found 541.2173.

3l: HRMS (ESI) m/z: [M + Na]⁺ Calcd for C₁₃H₂₂O₂Na 233.1512; Found 233.1505.

d. ¹H NMR spectra should consistently be listed with three significant digits. Compounds **4m**, **4v**, and **4ae** have only two significant digits.

Reply: Thank you for your professional suggestion. We have addressed the above issues in supporting information.

6. This manuscript and Supplementary Information have many errors and typos. Some of them are listed below.

a. Page 1, the 6th line from the bottom. "chen's group" -> "Chen's group"

Reply: Thanks for your correction. "chen's group" was corrected to "Chen's group" (Manuscript page 1, line 12).

b. Page 2, Fig. 1 (both a figure and a legend). "Isoquinolin" -> "Isoquinoline"

Reply: Thanks for your correction. "Isoquinolin" was corrected to "Isoquinoline" (Manuscript page 2, Figure 1b and legend).

c. Page 2, the 9th and 10th lines from the bottom. "Stille couplings (Fig. 2b) and the radical-radical homocouplings (Fig.2c) of glycols" -> "Stille couplings (Fig. 2b) of glycols and the radical-radical homocouplings (Fig.2c) of glycosides" (i.e., Substrates in Fig.2c are not glycols but glycosides)

Reply: Thanks for your correction. The sentence "Stille couplings (Fig. 2b) and the radical-radical homocouplings (Fig.2c) of glycols." was corrected to "Stille couplings (Fig. 2b) of glycols and the radical-radical homocouplings (Fig.2c) of glycosides." In addition, we also corrected the errors in the subheadings of Fig. 2b and Fig. 2c in Figure 2. (Manuscript page 2, line 15-16).

d. Page 3, Fig. 2c, the chemical structure of acyl telluride. "PhTeCO" -> "COTePh"

Reply: Thanks for your correction. "PhTeCO" was corrected to "COTePh" in Fig. 2c.

e. Page 3, Fig. 2d. The glycol of the Pd(IV) intermediate is missing the letter "n" near the parenthesis.

Reply: We are sorry for our negligence. The letter "n" was added in Fig. 2d.

f. Page 4, the 7th line of Results and discussion section. "ethyl acrylate." is unnecessary.

Reply: We are sorry for our negligence. "ethyl acrylate." was deleted.

g. Page 4, the 14th line of Results and discussion section. "entries 11-13" -> "entries 8, 11-13"

Reply: Thanks for your correction. "entries 11-13" was changed to "entries 8, 11-13" (Manuscript page 4, line 20).

h. Page 4, the 4th line from the bottom. "smNBEs" -> "smNBE"

Reply: Thanks for your correction. "smNBEs" was corrected to "smNBE" (Manuscript page 4, the 6th line from the bottom).

i. Page 5, Table 1, the chemical structure of **N₄**. The ester moiety should be oriented in the same plane as olefins.

Reply: Thanks for your correction. **N₄** was corrected.

j. Page 5, the 2nd line of Substrate scope section. “glycoacyl” -> “glycosyl”

Reply: Thanks for your correction. “glycoacyl” was corrected to “glycosyl” (Manuscript page 5, line 2 Substrate scope section).

k. Page 7, line 4. “acetyl (OAc)” -> “acetyl (Ac)”

Reply: Thanks for your correction. “acetyl (OAc)” was corrected to “acetyl (Ac)” (Manuscript page 7, line 1).

l. Page 8, the 1st line of Synthetic utility section. “Thereafte” -> “Thereafter”

Reply: Thanks for your correction. “Thereafte” was corrected to “Thereafter” (Manuscript page 8, line 1 Synthetic utility section).

m. Page 8, the 4th line of Synthetic utility section. “(triisopropylsilyl)acetylene” -> “zinc cyanide”

Reply: Thanks for your correction. “(triisopropylsilyl)acetylene” was corrected to “zinc cyanide” (Manuscript page 8, 4th line of Synthetic utility section).

n. page 10, reference 8 “(2022)” -> “(2023)”

Reply: Thanks for your correction. “(2022)” was corrected to “(2023)” (Manuscript Page 10, reference 8)

o. Page 11, reference 57. The citation lacks the journal name “Angew. Chem. Int. Ed.”.

Reply: Thanks for your correction. The journal name “Angew. Chem. Int. Ed.” added to references. (Manuscript Page 14 reference 55)

p. Page S1, the 6th line of Instrumentation section. The “3” of “CDCl₃” should be a subscript.

Reply: Thanks for your correction. The “3” of “CDCl₃” has been changed to a subscript.

Reviewer #3 (Remarks to the Author):

The authors present a new method to synthesize C-oligosaccharides through Catellani-type three-component coupling involving 2-iodoglycals and glycosyl chlorides. The developed method is remarkable because it stereoselectively provides various C-oligosaccharides in a single operation. Moreover, several C-oligosaccharide conjugate molecules were synthesized using this method. **Therefore, this work is worthy to publish**, but the manuscript and Supporting Information should be thoroughly revised and resubmitted because scholarly descriptions are inappropriate. Additional comments are provided below.

Reply: We thank the reviewer for the positive recommendation. Much appreciated! We carefully checked the entire manuscript and supporting information, and corrected any incorrect expressions in the manuscript and supporting information to make the article more readable.

(1) L24, “thusas”: Typo?

Reply: Thank you for your professional suggestion. The sentence “monofunctionalization **thuas** far typically involved Heck.” was changed to “monofunctionalization typically involves palladium-catalyzed Heck.” (Manuscript page 1, line 6-7).

(2) L38-41, “C-oligosaccharides are pharmacologically relevant compounds ~ , rendering them preferred chemotherapy drugs, such as maitotoxin, /~”: Maitotoxin is a poisonous compound rather than a drug. The descriptions on maitotoxin should be removed from the main text and Fig. 2. Ref 29 should also be deleted because it describes the total synthesis of hikizimycin.

Reply: Thank you for your valuable suggestions. We carefully considered “maitotoxin” and realized that

we had written inappropriately. The sentence “C-oligosaccharides are pharmacologically relevant compounds as they are more resistant to enzymatic degradation and chemical hydrolysis than are O-oligosaccharides, rendering them preferred chemotherapy drugs, such as maitotoxin,²⁸⁻²⁹ dodecodiulose,³⁰ tunicamine (Fig. 2a).³¹” was corrected to “Compared with O-oligosaccharides, C-oligosaccharides have remarkable stability in the process of enzyme and chemical hydrolysis, and are also widely present in natural products and drug molecules. For example, marine natural product (maitotoxin)²⁸ and drug molecules (dodecodiulose and tunicamine)²⁹⁻³⁰ featuring interglycosidic C-linkages (Fig. 2a).” (Manuscript page 2, line 1-4).

Ref 29 has been deleted.

(3) L45: Ref 35 is inappropriate because a natural product synthesis rather than a C-oligosaccharide synthesis was reported.

Reply: We appreciate this comment. “hetero-Diels-Alder cycloaddition³⁵” and Ref 35 have been deleted.

(4) L48, “C-at two anomeric carbons”: This part is unclear.

Reply: Thank you for your valuable suggestion. The sentence “In addition, methods that directly provide access to **C-at two anomeric carbons** are rare.” was changed to “In addition, the methods for assembling C-disaccharides that involve directly connecting the anomeric carbon and anomeric carbon are rare.” (Manuscript page 2, line 11-12).

(5) L50, “polarity reversal at the anomeric”: Typo?

Reply: Thank you for your valuable suggestion. “polarity reversal at the anomeric” was changed to “anomer carbon polarity reversal.” (Manuscript page 2, line 13-14).

(6) L52-53, “and have limited substrate scope (Fig. 2a).”: Fig, 2a shows some bioactive compounds rather than the reaction scope.

Reply: Thanks for your correction. “(Fig. 2a)” has been deleted.

(7) L81-82, “As shown in Table 1, ~conditions for the envisioned palladium/norbornene (NBE)-enabled ~ via cooperative catalysis ~”: “palladium/norbornene (NBE)-enabled” and “via cooperative catalysis” are almost the same. Delete the latter.

Reply: Thank you for your valuable suggestion. “via cooperative catalysis” has been deleted.

(8) L87, “bases.ethyl acrylate”: Typo

Reply: We are sorry for our negligence. “ethyl acrylate.” was deleted.

(9) L95: Is “Ph-DavePhos” L1? Indicate clearly.

Reply: Thank you for your valuable suggestions. “Ph-DavePhos” is **L1** and has been indicated (Manuscript page 4, line 22)

(10) L100: What is “ANP”?

Reply: Thank you for your professional suggestion. “ANP” is “alkenyl-norbornadiene-palladacycle”. It has been indicated in the abstract.

(11) L103: The reason why N₈ is optimal should be provided (see below).

Reply: We appreciate this suggestion. Please see comment 21 below.

(12) L111, “2-iodioses”: Typo?

Reply: Thanks for your correction. “2-iodioses” was corrected to “2-iodoglycals.” (Page 8, line 2 of Substrate scope.)

(13) L117, "ethyl tert-butyl": typo?

Reply: Thanks for your correction. "ethyl tert-butyl" was corrected to "tert-butyl acrylate." (Page 5, line 8 of Substrate scope.)

(14) L125, (4h-4k)": typo? "4i-4k" is correct?

Reply: The sentence "In addition to D-2-iodoglucal, the reactions of glycols **1i-1k**, derived from D-galactose and L-rhamnose, with glycosyl chloride donor **2a** and acrylate **3a** were also compatible under the reaction conditions (**4h-4k**)." Next, we examined the scope of 2-iodoglycols (derived from D-galactose and L-rhamnose), and the reactions of 2-iodoglycols **1c**, **1d**, and **1e** with **2a** and **3a** were gave target products **4i**, **4j**, and **4k** with yields of 59%, 64%, and 50%, respectively." (Manuscript page 6, line 1-3)."

(15) Fig. 4A: "Gram-scale synthesis" is incorrect because the amount of starting 2-iodoglucal is less than 1 g and 471.1 mg of **4a** was obtained. The label "Synthetic amide" is also strange.

Reply: Thanks for your corrections. We did the gram-scale synthesis again. (Manuscript page 7, Fig. 4 and Supporting Information page 12)."

"Synthetic amide" was changed to "Conversion to amide." Manuscript page 7, Fig. 4)

(16) L141-142, "D-ribofuranose (**4v**), D-glucose (**4w**), D-galactose (**4x**), D-mannose (**4y**) as terminating reagents,": This description is strange because **4v-4y** are the products rather than terminating agents. "D-ribofuranose-, D-glucose-, D-galactose-, and D-mannose-derived acrylates" are the terminating agents.

Reply: Thanks for your correction. The sentence "High-efficiency diastereoselective one-pot synthesis of trisaccharides was conducted using D-ribofuranose (**4v**), D-glucose (**4w**), D-galactose (**4x**), D-mannose (**4y**) as terminating reagents with three-component reactions of 2-iodoglycol **1a** and glycosyl chloride donor **2a**." was corrected to "One pot efficient diastereoselective synthesis of trisaccharides were carried out using D-ribofuranose (**3h**), D-glucose (**3i**), D-galactose (**3j**), D-mannose (**3k**) as terminating reagents." (Manuscript page 7, line 10-12).

(17) L153, "Thereafte", "succsfully": Typos?

Reply: Thanks for your correction. "Thereafte" was corrected to "Thereafter". "succsfully" was corrected to "successfully." (Manuscript page 8, line 1 Synthetic utility section).

(18) L161, (Fig.4A): typo? "Fig. 4B" is correct?

Reply: Thanks for your correction. We find that the position of "(Fig.4A)" is not correct. "(Fig.4A)" was changed to the previous sentence. (Manuscript page 8, line 4 of Synthetic utility section).

(19) L173-174, "the total energy barrier is -25.7 kcal mol⁻¹": This description is very strange. What does an energy barrier with a negative value mean?

Reply: Thanks for your correction. The total energy barrier required 24.3 kcal/mol from **1h** to **D** for **N₈** after the solvation free energies were included, and we revised the value in the manuscript. (Manuscript page 8, line 6 of Mechanistic investigation section).

(20) L179-180: Figure S2 shows the IRC result for OA through **4-TS** rather than that for RE through **5-TS**.

Reply: Thanks for your professional comment. The combinations of Figure S4 and Figure S5 are to explain that the oxidative addition reaction is not through the oxocarbenium intermediate, but through a three-membered transition state. We rewrote the sentence "The DFT calculations revealed that the OA proceeds via a concerted three-membered cyclic transition state (**4-TS**), which directly leads to form the D-mannofuranosyl Pd(IV) intermediate (F- α) without the formation of oxocarbenium^{26, 56} (see the

Supporting Information for intrinsic reaction coordinate (IRC) calculations that prove the nature of this transition state).” (Manuscript page 8, line 11-12 of Mechanistic investigation section). In addition, the dissociation energy of the corresponding glycosylation from the Pd(IV) species was also calculated. The Pd-C1 bond of the Pd(IV) species (**F- α**) dissociates to form the oxocarbenium ion with absorbing 42.4 kcal/mol, which hints that the Pd(IV) intermediate is configurationally stable (*CCS Chem.* **2020**, *2*, 1729). The IRC for **5-TS** will show the C-C forming process.

(21) DFT calculations using NBE should give incorrect estimations of energy surfaces because NBE is much less efficient than **N₈**, which is experimentally used. Therefore, DFT calculations should be conducted using **N₈**. Compare results obtained from DFT calculations using other NBEs to discuss why **N₈** is optimal.

Reply: Thanks for your professional comment. DFT calculations have been changed to use **N₈** for calculation (See manuscript and Supporting Information). We compared the migratory insertion transition states for different NBEs (i.e. **N₁-N₃**, **N₅**, **N₈**) from **B** to **2-TS**. The calculated results are listed as below:

All energies are with respect to B and in kcal/mol.

From the Figure, it could be found that the total energy barriers of **2-TS** (**N₈**, **N₅**, **N₂**, **N₁** and **N₃**) are 24.7, 27.5, 27.8, 28.2 and 29.8 kcal/mol, respectively. The computational results are in agreement with the experimental observations.

In general, C5-substituted smNBEs exhibit reactivity similar to that of simple unsubstituted NBE, and the exact reason is unclear (*J. Am. Chem. Soc.* **2020**, *142*, 17859–17875; *J. Am. Chem. Soc.* **2020**, *142*, 3050–3059; *Chem. Rev.* **2019**, *119*, 7478–752). The steric and electronic environment of the NBE alkene in these smNBEs is almost identical to that of simple NBE, though the remote steric effect or coordinative properties of the C5 substituents could play an important role in modulating reaction selectivity. In our reaction, we speculate that the possible reason is that the electronic effects of the C5 substituents play an important role in the reaction. The C5-substituted smNBEs (-OCOCH₃ for **N₈**, -COOCH₃ for **N₅**, -CN for **N₂**, and -COCH₃ for **N₃**) could have electrostatic attractions with one Cs atom which is denoted by O(N)-Cs distances and could lower the barrier. However, the smNBEs could have repulsive interactions with the neighbor Cs₂ICO₃ group which is denoted by C-Cs distances and could elevate the barrier. For **N₈**, it has the shortest O-Cs distance and the longest C-Cs distances, which indicates that it has the strongest attractions and the weakest repulsions, and therefore has the lowest barrier and the optimal reactivity (see below). On the other hand, we cannot rule out whether there are other spatial effects or coordination

properties affecting the activity of C5-substituted smNBEs. We really hope to provide a more detailed explanation of the activity of C5 substituted smNBEs from remote steric effect or coordinative properties in our laboratory in the future.

Migratory-insertion transition states with N₅, N₅, N₂, N₁ and N₃. Bond distances are in Å.

(22) L216: "n/a" => "62"

Reply: Thanks for your correction. "n/a" was corrected to "62". (Manuscript Page 10, reference 3)

(23) SI: Are starting materials known compounds? If so, references, which report full characterization data, should be cited for each compound. For new compounds, MS data should be provided.

Reply: Thank you for your professional suggestion. For known starting materials, we have provided references, and for unknown starting materials, in addition to providing synthesis methods, we have also provided spectra and mass spectrometry data of the products in the supporting information.

(24) The deviation of the MS data for **4m** from calcd value is over 0.0030.

Reply: Thank you for your professional suggestion. We retested the high resolution mass spectrometry and found molecular mass with lower relative error values as follows:

4m: HRMS (ESI) m/z: [M + H]⁺ Calcd for C₇₅H₈₁O₁₁Si 1185.5543; Found 1185.5525.

(25) The deviation of the MS data for **4ac** from calcd value is over 0.0030.

Reply: Thank you for your professional suggestion. We retested the high resolution mass spectrometry and found molecular mass with lower relative error values as follows:

4ac: HRMS (ESI) m/z: [M + H]⁺ Calcd for C₆₀H₆₉O₁₂ 981.4784; Found 981.4788.

(26) The deviation of the MS data for **4ad** from calcd value is over 0.0030.

Reply: Thank you for your professional suggestion. We retested the high resolution mass spectrometry and found molecular mass with lower relative error values as follows:

4ad: HRMS (ESI) m/z: [M + K]⁺ Calcd for C₆₂H₇₈KO₁₂ 1053.5125; Found 1053.5103.

(27) The deviation of the MS data for **8a** from calcd value is over 0.0030.

Reply: Thank you for your professional suggestion. We retested the high resolution mass spectrometry and found molecular mass with lower relative error values as follows:

8a: HRMS (ESI) m/z: [M+H]⁺ Calcd for C₅₆H₆₆NO₁₃ 960.4529; Found 960.4532

REVIEWERS' COMMENTS

Reviewer #2 (Remarks to the Author):

The revised version mostly addressed the concerns that were raised by the three referees. The results and discussion are clearly presented. Accordingly, the scientific level of the manuscript is significantly improved. I recommend publishing the paper in Nature Communications after the amendment of the following point.

1. Fig. 5a

The structures need to be enlarged.

2. Figure S1

The structure of D is not correct and needs to be replaced.

Reviewer #3 (Remarks to the Author):

Points suggested in the first-round review process have been mostly addressed in the revised manuscript. However, there are still several points to be revised. Therefore, I would like to suggest the manuscript should be revised according to the following comments, before the final acceptance.

(1) Caption of Fig 2a: Add “and drug molecules” after “Natural products”.

(2) L69-73: Periods before “2” and “3” should be replaced by commas.

(3) L93: “the role of each reactions” should be corrected as “the role of each reaction components”.

(4) L119: “amide, ketone, and pyridine” should be corrected as “acrylamide, vinyl ketone, and 2-vinylpyridine”.

(5) L127: “were gave” Typo?

(6) L155: “cooperateive” Typo?

(7) L160: “Desiliconized” should be corrected as “desilylated”.

(8) “Mechanistic investigation” section: Whole Figure S1 should be included in the main manuscript.

(9) L172: What does “alkenyl-norbornadiene-palladacycle” mean?

(10) L173: “24.3 kcal/mol”? 24.7 kcal/mol is correct?

(11) L186: “the F-beta required an energy barrier of 28.8 kcal/mol higher in energy than F-alpha” is very strange. The authors might mean that “F-beta is located 28.8 kcal/mol higher in energy than F-alpha”.

REVIEWERS' COMMENTS

Reviewer #2 (Remarks to the Author):

The revised version mostly addressed the concerns that were raised by the three referees. The results and discussion are clearly presented. Accordingly, the scientific level of the manuscript is significantly improved. I recommend publishing the paper in Nature Communications after the amendment of the following point.

Reply: We thank the reviewer for the positive recommendation. Much appreciated!

1. Fig. 5a. The structures need to be enlarged.

Reply: Thank you for your professional suggestion. The structures in Figure 5a and Figure S1 have been enlarged in manuscript and supporting information, respectively.

2. Figure S1. The structure of D is not correct and needs to be replaced.

Reply: Thanks for your correction. The structure of D was corrected and replaced in supporting information.

Reviewer #3 (Remarks to the Author):

Points suggested in the first-round review process have been mostly addressed in the revised manuscript. However, there are still several points to be revised. Therefore, I would like to suggest the manuscript should be revised according to the following comments, before the final acceptance.

Reply: We thank the reviewer for the positive recommendation. Much appreciated!

(1) Caption of Fig 2a: Add "and drug molecules" after "Natural products".

Reply: Thank you for your professional suggestion. After "Natural products" has been added "and drug molecules". (Manuscript, Caption of Fig 2a).

(2) L69-73: Periods before "2)" and "3)" should be replaced by commas.

Reply: Thank you for your professional suggestion. Periods before "2)" and "3)" have been replaced by commas. (Page 3, paragraph 1, lines 5-7).

(3) L93: "the role of each reactions" should be corrected as "the role of each reaction components".

Reply: Thank you for your professional suggestion. "the role of each reactions" has been corrected as "the role of each reaction components". (Manuscript, "Results" section, line 12).

(4) L119: "amide, ketone, and pyridine" should be corrected as "acrylamide, vinyl ketone, and 2-vinylpyridine".

Reply: Thank you for your professional suggestion. "amide, ketone, and pyridine" have been corrected as "acrylamide, vinyl ketone, and 2-vinylpyridine". (Manuscript, "Substrate scope" section, lines 8-9).

(5) L127: "were gave" Typo?

Reply: Thanks for your correction. "were gave" was corrected to "resulted in the". (Manuscript, Page 8, paragraph 1, line 1).

(6) L155: "cooperateive" Typo?

Reply: Thanks for your correction. "cooperateive" was corrected to "cooperative". (Manuscript, "Synthetic utility" section, line 1).

(7) L160: "Desiliconized" should be corrected as "desilylated".

Reply: Thanks for your correction. “Desiliconized” was corrected to “desilylated”. (Manuscript, “Synthetic utility” section, line 6).

(8) “Mechanistic investigation” section: Whole Figure S1 should be included in the main manuscript.

Reply: Thank you for your professional suggestion. We added whole Figure S1 to the manuscript. (Manuscript, “Mechanistic investigation” section).

(9) L172: What does “alkenyl-norbornadiene-palladacycle” mean?

Reply: We apologize for the incorrect expression. “alkenyl-norbornadiene-palladacycle” was corrected to “alkenyl-norbornyl-palladacycle”. In the Catellani reaction, aryl-norbornyl-palladacycle (**ANP**) is an important five-membered palladacycle intermediate obtained through a series of reactions between the aryl substrate and norbornene (*J. Am. Chem. Soc.* **2011**, *133*, 22, 8574–8585; *Chem. Rev.* **2019**, *119*, 12, 7478–7528). Alkenyl-norbornyl-palladacycle (**ANP**) is also an important five-membered palladacycle intermediate obtained through a series of reactions between the alkenyl substrate and norbornene (*Nat. Chem.* **2019**, *11*, 1106–1112).

(10) L173: “24.3 kcal/mol”? 24.7 kcal/mol is correct?

Reply: We apologize for the unclear expression. What we want to express is that the energy barrier **1h** to **D** required 24.3 kcal/mol, rather than **B** to **2-TS** required 24.7 kcal/mol. Therefore, we have described it appropriately as follows. Our calculated results showed that the energy barrier from **1h** to **D** require 24.3 kcal/mol, which include a series of processes such as the migratory insertion of norbornene and subsequently C1-H activation to form alkenyl-norbornyl-palladacycle (**D**). (Manuscript page 9, line 3-6, Mechanistic investigation section).

(11) L186: “the F-beta required an energy barrier of 28.8 kcal/mol higher in energy than F-alpha” is very strange. The authors might mean that “F-beta is located 28.8 kcal/mol higher in energy than F-alpha”.

Reply: We quite agree with your opinion and apologize for our inappropriate expression. The sentence “the F-beta required an energy barrier of 28.8 kcal/mol higher in energy than F-alpha” was corrected to “**F-β** is located 28.8 kcal/mol higher in energy than **F-α**”. (Manuscript page 10, Last line, Mechanistic investigation section).